# Systemic Steroids in Preventing Bronchopulmonary Dysplasia (BPD): Neurodevelopmental Outcome According to the Risk of BPD in the EPICE Cohort

**DOI:** 10.3390/ijerph19095600

**Published:** 2022-05-05

**Authors:** Noura Zayat, Patrick Truffert, Elodie Drumez, Alain Duhamel, Julien Labreuche, Michael Zemlin, David Milligan, Rolf F. Maier, Pierre-Henri Jarreau, Héloïse Torchin, Jennifer Zeitlin, Alexandra Nuytten

**Affiliations:** 1Department of Neonatology, University Hospital of Nantes, F-44093 Nantes, France; 2ULR 2694—METRICS, Évaluation des Technologies de Santé et des Pratiques Médicales, University Lille, CHU Lille, F-59000 Lille, France; patrick.truffert@chu-lille.fr (P.T.); edrumezchr@gmail.com (E.D.); alain.duhamel@univ-lille2.fr (A.D.); julien.labreuche.chru@gmail.com (J.L.); alexandra.nuytten@chu-lille.fr (A.N.); 3Department of Neonatology, Jeanne de Flandre University Hospital, F-59000 Lille, France; 4Department of Biostatistics, CHU Lille, F-59000 Lille, France; 5Department of General Paediatrics and Neonatology, Medical School, Saarland University, G-66421 Homburg, Germany; michael.zemlin@uks.eu; 6Faculty of Medical Science, Newcastle University, Newcastle upon Tyne UK-NE1 4LP, UK; dwamilligan@gmail.com; 7Children’s Hospital, University Hospital, Philipps University Marburg, G-35043 Marburg, Germany; rolf.maier@med.uni-marburg.de; 8Neonatal Intensive Care Unit of Port-Royal, Cochin Hospital, APHP, F-75014 Paris, France; pierre-henri.jarreau@aphp.fr (P.-H.J.); heloise.torchin@inserm.fr (H.T.); 9CRESS, Obstetrical Perinatal and Paediatric Epidemiology Research Team, EPOPé, INSERM, INRA, University of Paris, F-75004 Paris, France; jennifer.zeitlin@inserm.fr

**Keywords:** neurodevelopmental outcome, bronchopulmonary dysplasia, preterm birth, post natal steroid therapy

## Abstract

Background: Postnatal steroids (PNS) have been used to prevent bronchopulmonary dysplasia (BPD) in preterm infants but have potential adverse effects on neurodevelopment. These effects might be modulated by their risk of BPD. We aimed to compare patients’ neurodevelopment with PNS treatment according to their risk of BPD in a European cohort. Methods: We developed a prediction model for BPD to classify infants born between 24 + 0 and 29 + 6 weeks of gestation in three groups and compared patients’ neurological outcome at two years of corrected age using the propensity score (PS) method. Results: Of 3662 neonates included in the analysis, 901 (24.6%) were diagnosed with BPD. Our prediction model for BPD had an area under the ROC curve of 0.82. In the group with the highest risk of developing BPD, PNS were associated with an increased risk of gross motor impairment: OR of 1.95 after IPTW adjustment (95% CI 1.18 to 3.24, *p* = 0.010). This difference existed regardless of the type of steroid used. However, there was an increased risk of cognitive anomalies for patients treated with dexa/betamethasone that was no longer observed with hydrocortisone. Conclusions: This study suggests that PNS might be associated with an increased risk of gross motor impairment regardless of the group risk for BPD. Further randomised controlled trials exploring the use of PNS to prevent BPD should include a risk-based evaluation of neurodevelopmental outcomes. This observation still needs to be confirmed in a randomised controlled trial.

## 1. Introduction

Bronchopulmonary dysplasia (BPD) is a chronic lung disease that affects preterm infants and contributes to their morbidity and mortality. The NICHD defines BPD as any need for supplemental oxygen for more than 28 days postnatal age, and its severity depends on the respiratory status at 36 weeks post menstrual age (PMA) [1]. Postnatal steroids (PNS) have been shown to prevent progression to BPD and facilitate the weaning of ventilator-dependent infants [2]. However, they are also associated with an increased incidence of neurodevelopmental impairments (NDIs) [3,4]. Therefore, when considering PNS, the neonatologist has to balance the respiratory benefits and the neurological risks of PNS knowing that BPD itself increases the risk of neurodevelopmental impairments (NDIs) [5]. Different molecules such as dexamethasone, betamethasone or hydrocortisone can be prescribed. Hydrocortisone is now used preferentially as it has shown efficacy against BPD without significant difference in NDIs at 2 years corrected age [6]. In a meta-analysis of randomised control trials (RCTs), Doyle and al suggested that for patients at high risk of BPD, PNS could be beneficial to neurodevelopment. They posited that the benefits associated with preventing BPD using PNS could outweigh the adverse neurological effects of PNS in this specific sub-group [7]. They found that when the rate of dysplasia in the control group was greater than 65%, the group treated with PNS had lower risks of BPD as well as cerebral palsy (CP) or death. In our study, we aimed to test Doyle’s hypothesis in real-life conditions using a large cohort of preterm infants. Our objective was to evaluate the association between PNS and NDIs according to the risk of BPD, supposing that for a patient at high risk of BPD, treatment would improve both respiratory and neurological outcomes. We conducted a secondary analysis describing the effects on neurodevelopment according to the type of steroids.

## 2. Materials and Methods

### 2.1. Study Population

The EPICE (Effective Perinatal Intensive Care in Europe) cohort was designed to assess unit practices and children’s development across Europe. It included patients born between 22 + 0 and 31 + 6 weeks GA in 19 regions of 11 European countries in 2011/2012 [8]. Data from medical records were collected prospectively over varying 12 months periods (6 months for France). Investigators used a pretested standardized questionnaire jointly revised by the EPICE research group. Infants were followed up until discharge from the hospital or into long-term care or death. At 2 years of corrected age, questionnaires were sent to the parents to collect information on longer-term health outcomes. Ethics approval was obtained in each study region from regional and/or hospital ethics committees, as required by national legislation. The study was also approved by the French Advisory Committee on the Use of Health Data in Medical Research (CCTIRS) and the French National Commission for Data Protection and Liberties (CNIL). More information about data access can be found on the EPICE website (www.epiceproject.eu, accessed on 1 May 2022). We restricted our analyses to patients born between 24 + 0 and 29 + 6 GA admitted in a Neonatal Intensive Care Unit (NICU) because of the low BPD prevalence beyond 30 weeks GA and the small number of patients before 24 weeks [9].

### 2.2. Post-Natal Steroids and Bronchopulmonary Definitions

Post-natal steroids referred to systemic corticosteroids used to prevent BPD. The patients received either betamethasone, dexamethasone, hydrocortisone or a mixed exposure. The time of initiation, dose and length of the treatment depended on each unit’s protocol. In the secondary analysis, we grouped patients treated with betamethasone and dexamethasone as the molecules use similar pathways. BPD was defined according to the NIH [10] as dependence on supplementary oxygen or respiratory support, including continuous positive airway pressure or mechanical ventilation, at the age of 36 weeks post-menstrual age. Due to the design of the cohort, an oxygen reduction test could not be performed for all infants with mild oxygen supplementation at 36 weeks.

### 2.3. Classification According to the Risk of BPD

To classify our population according to their risk of developing BPD, we created our own model because the existing prediction models for BPD [11] used variables that were not all available in the EPICE questionnaires. We chose variables that were clinically relevant or that had been shown to be associated with BPD in the literature. The variables were all already available in the two first weeks of life [12] and infants who died before the age of 14 days were excluded from the analysis.

A total of 15 variables were pre-selected: gestational age in weeks defined as the best obstetric assessment based on last menstrual period and antenatal ultrasounds, sex, small for gestational age (SGA) defined as a birth weight ≤3rd percentile according to intrauterine growth curves, multiple pregnancies, inborn birth referring to infants being born in a centre with a NICU able to provide neonatal care in the first 48 h, hypertensive disorders of pregnancy (pre-eclampsia, HELLP syndrome or eclampsia), premature rupture of membranes (PROM), antenatal steroids, Apgar score below 7 at 5 min, surfactant therapy, respiratory support in the first 48 h of life, use of non-steroidal anti-inflammatory drugs (NSAID) treatment before day 3 for a persistent ductus arteriosus (PDA), early onset sepsis (EOS) defined as clinical signs of infection and a positive bacterial or fungal culture within the first 72 h of life and necrotizing enterocolitis (NEC) in the first 14 days of life. Invasive respiratory support was defined as a binary variable that referred to the use of mechanical ventilation immediately after birth and for more than 24 h or within the first two days of life.

### 2.4. Neurodevelopmental Outcome

Neurodevelopmental impairment was defined by abnormal gross motor function or cognitive development at two years of corrected age. Gross motor impairment was assessed according to parent-reported motor function with a questionnaire written in English for the study and translated into national languages. The answer “no” to one of the items defined gross motor impairment. To assess abnormal cognitive development we used two validated parental questionnaires: the third edition of the Ages and Stages Questionnaire (ASQ) for French regions and the Parent Report of Children Abilities–Revised (PARCA-R) for the rest of Europe [13,14]. These two questionnaires showed good performances to detect abnormal cognitive development compared with the revised Brunet Lezine psychometric test and the Mental Development Index of the Bayley Scales of Infant development III respectively. Cognitive impairment was defined by a non-verbal PARCA score <22 or an ASQ-3 problem-solving score beyond 2 standard deviations (SDs) below the mean or a vocabulary counting of fewer than 10 words. We repeated the analysis in the subgroup treated with hydrocortisone.

### 2.5. Statistical Analysis

In order to classify at day 14 infants into three tertiles of risk of BPD, we developed a predictive model using a backward-stepwise multivariable mixed logistic regression model with the center as a random effect (see Appendix A). Infants were classified into three BPD risk groups by using the tertiles of probability estimated from the predictive model. We compared neurodevelopment at 2 years corrected age in the group treated PNS compared with the control group in the overall study population and in each BPD risk group using mixed logistic regression models; odds ratios (ORs) and their 95% confidence intervals (CIs) were calculated as effect sizes using untreated infants as reference [8,15]. Because children were lost to follow-up, we compared their characteristics to the included population. Previous analyses of this cohort have illustrated the minimal impact of attrition on rates of neurodevelopment. We performed a subgroup analysis to differentiate patients treated with hydrocortisone. To take into account the potential indication bias, neurodevelopmental outcomes were compared using the inverse probability of treatment weighting (IPTW) propensity score (PS) method. The PS was estimated using a multivariable logistic regression model, with PNS-treatment as the dependent variable and all potential confounders selected on subject-matter knowledge as independent variables. Missing data for all the variables used in the analyses and on the outcome were imputed by multiple imputations under the missing-at-random assumption. Full details are available in the Appendix A. Statistical testing was conducted at the two-tailed α-level of 0.05. Data were analysed using Rstudio (Version 1.1.456) and the SAS version 9.4 (SAS Institute, Cary, NC, USA) software.

## 3. Results

### 3.1. Population Characteristics and Prediction of the Risk of BPD

Among 10,329 preterm infants included in the EPICE cohort from April 2011 to June 2012, 3662 infants were eligible for the study (Figure 1). The 434 patients who died before day 14 were born earlier, had a smaller weight and more Apgar scores <7 at 5 min of life, more invasive ventilatory support, high-grade (III or IV) intraventricular hemorrhage (IVH) and EOS than infants included in the study (Appendix A). BPD was diagnosed in 901 patients (24.6%, 95% CI [23–26%]). The six independent predictors included in our risk model for BPD were: gestational age, small for gestational age, male sex, surfactant use, invasive initial respiratory support and a PDA (Table 1). The performances of the model were good in terms of discrimination (median c-statistic from the 10 imputed datasets of 0.82, ranging from 0.819 to 0.823) and calibration (Appendix A). There were 1211, 1228 and 1223 patients in the high, moderate and low-risk group and 582 (48.1%), 259 (21.1%) and 69 (5.6%) patients with BPD, respectively. In the high-risk group, 344 (28.4%) patients were treated with PNS compared with 157 (12.8%) in the moderate risk group and 57 (3.8%) in the low-risk group. Dexamethasone was the most prescribed steroid: 207 (37.1%) versus 182 (32.6%) for hydrocortisone and 81 (14.5%) for betamethasone. PNS dose, timing and length of use are detailed in the Appendix A (Appendix A). At two years of corrected age, patients classified in the highest group of risk were more often treated for respiratory decompensations (14.8% vs. 12.9% and 9.0%, *p* = 0.031).

### 3.2. Exposure to PNS and Neurodevelopmental Outcome

In the untreated group, 116 infants (3.9%) died during hospitalisation or follow-up by comparison to 55 (9.9%) among infants treated with PNS (*p* < 0.001). Infants exposed to PNS during hospitalisation in the NICU were more frequently boys, born before 28weeks GA, had more often invasive initial respiratory support and surfactant therapy, PDA requiring treatment and Apgar score lower than 7 at 5 min and had more often high-grade IVH (Appendix A). After IPTW adjustment, the mean standardised differences between groups were below 10% (see Appendix A). At two years of corrected age, 65.5% of the children were included in the follow-up and 1668 (49.0%) parents answered the PARCA or the ASQ questionnaires. The characteristics of the lost in follow-up population are presented in Appendix A. In the overall study population, PNS treatment was associated with an increased rate of gross motor impairment (12.1% in treated infants vs. 5.5%), with an unadjusted OR of 2.35 (95% CI, 1.69 to 3.26). However, after IPTW adjustment, this association was of borderline significance, with an IPTW-adjusted OR = 1.55 (95% CI, 0.99 to 2.41; Table 2). As shown in Figure 2, PNS treatment was associated with gross motor impairment in the highest BPD risk group (OR, 1.95; 95% CI, 1.18 to 3.24; *p* = 0.01). Heterogeneity in the association between PNS and motor or cognitive outcome by BPD risk was not significant (*p* = 0.13). Among the patients who received PNS, 34.1% had impaired cognitive development at 2 years compared with 25.2% in the untreated group. After IPTW adjustment, no significant difference was found in cognitive development in the global population as well as in the 3 groups at risk for BPD (Table 2, Figure 2).

In the hydrocortisone subgroup, treatment was still associated with gross motor impairment in the global population after IPTW adjustment but no longer in the different risk groups (Figure 3, Appendix A). The OR before adjustment was 3.29, 95% CI, 0.67 to 16.15). For patients treated with dexamethasone or betamethasone, findings were consistent with our main results with higher rates of motor impairment in the overall population as well as in the high-risk group. Moreover, the rates of cognitive impairment were higher in the treated group compared with the control group: OR, 2.16 (95% CI, 1.39 to 3.35) while there was no difference with hydrocortisone.

## 4. Discussion

For infants at high risk for BPD, PNS treatment was associated with higher rates of gross motor anomalies at two years of corrected age compared to the untreated group. This difference was observed regardless of the type of steroid. Furthermore, the dexamethasone or betamethasone treatment was associated with an additional risk of cognitive anomalies.

To our knowledge, this is the first study analysing in real practice conditions PNS effects on neurodevelopment according to the risk for BPD. Postnatal steroids prescription is heterogeneous in European NICUs [16] and Nuytten et al. showed that it depended more on each unit’s policy of prescription than on the patients’ respiratory severity. Additionally, units with a restricted PNS use policy didn’t have higher rates of BPD after adjustment (OR 0.68; 95% CI 0.45–1.03) [17]. Likewise, PNS treatment in our study was prescribed according to real-life practice and not necessarily to the most severe patients. Dexamethasone was the most prescribed steroid for respiratory indications in the EPICE cohort. Baud and al showed in 2005 that a neonatal injection of dexamethasone in mice strongly affected neuronal maturation, cortical GABAergic differentiation and induced apoptotic nonGABAergic neuron loss [18]. Its use has been linked to abnormal motor development as well as cognitive functions and behavioural disorders [4]. Instead of mitigating brain lesions, steroid use on high-risk patients might worsen neuronal injuries. Yet, in the randomised PREMILOC clinical trial published by Baud et al. in 2017, the early use of low-dose hydrocortisone improved survival without bronchopulmonary dysplasia in preterm infants born before 28 weeks GA with no significant difference in neurodevelopment at 2 years of age [19]. Yet, in our study, the effect of steroids on gross motor functions was similar in the hydrocortisone subgroup. Nevertheless, the negative effects of dexa/betamethasone on cognitive development were no longer observed with hydrocortisone. Inhaled budesonide is an alternative treatment with less exposure to adverse systemic steroids’ effects [20] and wasn’t evaluated in this study because of the low number of patients who received this therapy.

Thus, our findings, despite the observational design of the study, encourage us to follow up closely with the patients treated with PNS regardless of the type of steroid used, in particular on neurodevelopmental acquisitions. The main limitation of our work resides in its observational design with potential unknown factors interfering with patients’ neurological outcomes. To adjust our results to the variables associated with patients’ clinical severity, we used the PS method, a statistical method that recreates a pseudo randomisation on known confounders. Our results are consistent with the published literature on the subject as in Qin et al. study that showed similar results with postnatal dexamethasone exposure being associated with an increased proportion of neurological impairment [21].

In addition to the type of steroid or the treatment duration [22], the patient’s initial risk for BPD could influence neurodevelopmental outcomes. BPD is known to be a cause of NDI itself: repeated hypoxemia, hypercapnia and respiratory acidosis episodes lead to brain injuries and higher rates of cerebral palsy [23]. Reducing the severity of BPD was expected to outweigh its neurodevelopmental side effects [24]. Our results refute this hypothesis with an association between gross motor impairment and PNS, robust to PS adjustment. In the moderate or low-risk groups, no significant effect was observed which can be explained by the low number of treated patients and NDIs. No difference in cognitive development was shown between treated and untreated infants regardless of their risk group for BPD which is consistent with Crotty et al. study [25]. However, considering only dexa/betamethasone, treatment was associated in our study with higher rates of cognitive anomalies using PARCA-R and ASQ auto questionnaires. Both PARCA-R and the ASQ have been validated as screening tools and are used in follow-up consultations [26]. Even though around 1800 families replied and missing data were replaced using multiple imputations, extrapolation of our results on cognitive development should be read with some caution regarding the number of patients lost to follow-up.

Early prediction of survival without BPD was conducted by running a model based on clinical data at 14 days of age which, again, is a routine clinical practice condition. At 14 days, clinicians might start considering PNS treatment according to the patient’s respiratory severity. In the meta analyses published in 2005, Doyle et al. defined the risk of chronic lung disease as the rate of BPD in the control group [24]. Despite the fact that this is inapplicable in real life, this definition gives the same probability of BPD to a group of very different patients. It is therefore far from physicians’ reality that has to take each patient individually and in a very specific chronology. In 2011, Laughon et al. developed and validated a model that gave reliable probabilities for BPD or death [27] the best performances being achieved at 14 days of life. Because all of the variables required for this score weren’t included in the EPICE questionnaires, we elaborated our own classification model that was designed specifically for our population without external validation. Among the six selected predictors, gestational age, small for gestational age status and initial respiratory support contributed the most to improving the predictive capacity of our model. The same variables were selected by Laughon et al. and more recently by Baud et al. [28] in their predictive model for BPD [29]. All these studies remind us of the importance of including a risk-based strategy in future research on BPD to target a population that benefits the most from PNS even though, in our cohort treated mostly with dexamethasone, the high-risk group for BPD does not seem to benefit from PNS treatment. Nevertheless, upcoming randomised controlled trials exploring the use of postnatal steroids to prevent or treat BPD should include a “BDP risk” stratification strategy and long-term follow-up concerning neurodevelopmental outcomes.

## 5. Conclusions

Our study did not confirm a positive risk/benefit balance in high-risk BPD preterm newborns. On the contrary, postnatal steroid treatment in this group seems to be associated with a higher risk of motor impairment at two years of corrected age regardless of the type of steroid. Dexamethasone or betamethasone treatment was associated with an additional risk of cognitive anomalies. In the hydrocortisone subgroup, such association was no longer found. Randomized trials evaluating low-dose hydrocortisone and including a risk-based strategy are necessary to confirm these results.

## Figures and Tables

**Figure 1 ijerph-19-05600-f001:**
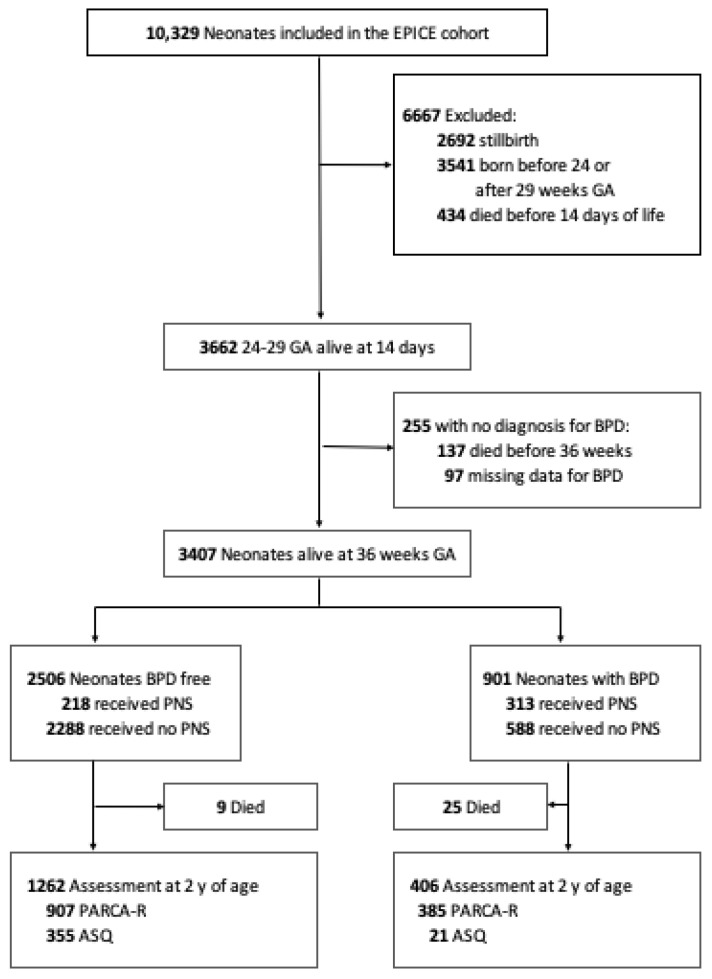
Study Flow Chart. Abbreviations: EPICE, Effective Perinatal Intensive Care in Europe; GA, gestational age; BPD, bronchopulmonary dysplasia; PNS, postnatal steroids; PARCA-R, Parent Report of Children Abilities–Revised; ASQ, Ages and Stages Questionnaire.

**Figure 2 ijerph-19-05600-f002:**
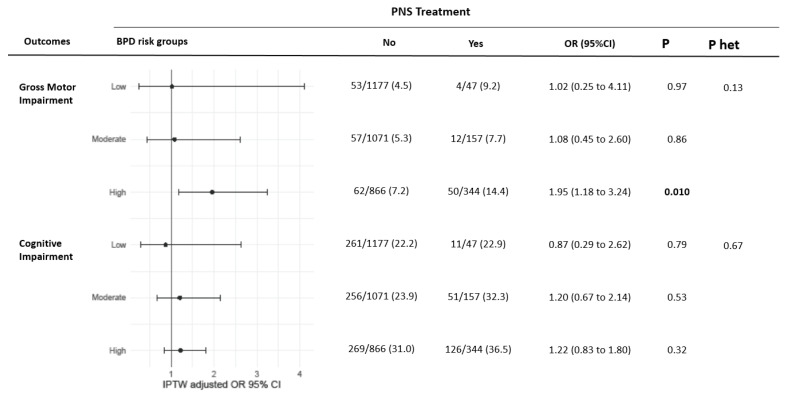
Association of PNS treatment with two-year neurodevelopmental outcomes according to BPD risk groups after IPTW adjustment. Values are reported as no./No. (%). P het indicates p for heterogeneity across BDP risk groups. Descriptive values, OR and *p*-value were calculated after handling missing values by multiple imputation. Abbreviations: IPTW, inverse probability of treatment weighting; OR, odds-ratio; CI, confidence interval.

**Figure 3 ijerph-19-05600-f003:**
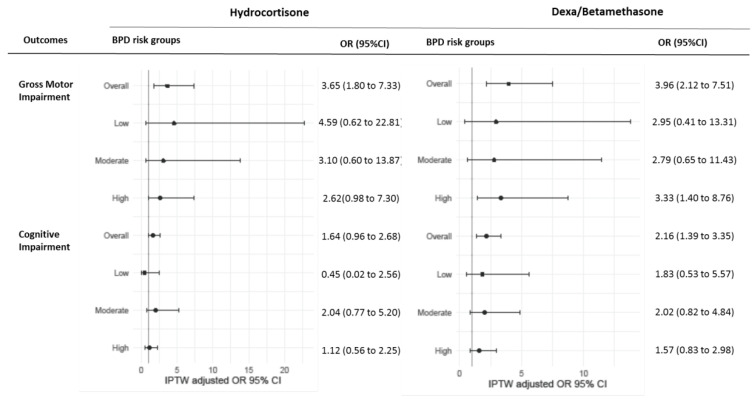
Association of PNS treatment with two-year neurodevelopmental outcome in patients treated with hydrocortisone and dexa/betamethasone after IPTW adjustment. Abbreviations: IPTW, inverse probability of treatment weighting; OR, odds-ratio; CI, confidence interval.

**Table 1 ijerph-19-05600-t001:** Independent predictors of Bronchopulmonary Dysplasia.

Predictors	OR (95% CI) ^1^	*p*-Value ^1^
Gestational age (weeks) ^2^	1.47 (1.39 to 1.56)	<0.001
SGA	3.21 (2.62 to 3.94)	<0.001
Male	1.51 (1.27 to 1.79)	<0.001
Surfactant	1.49 (1.14 to 1.94)	0.004
Initial respiratory support	2.61 (2.08 to 3.28)	<0.001
PDA requiring treatment	2.04 (1.32 to 3.15)	0.001

^1^ Estimated using a backward-stepwise mixed logistic regression model including center as random effect and after handling missing data by multiple imputation (m = 10). Candidate predictors were gestational age, SGA, born in a level 3-unit, inborn status, male gender, multiple pregnancy, prom, eclampsia, antenatal steroids injection, 5 min apgar score, surfactant therapy, early onset neonatal infection, PDA, initial respiratory support and necrotizing enterocolitis. ^2^ OR per 1-week decrease. Abbreviations: OR, odds-ratio; CI, confidence interval; SGA, small for gestational age; PDA, patent ductus arteriosus.

**Table 2 ijerph-19-05600-t002:** Comparison of two-year neurodevelopmental outcomes in PNS and non-PNS treated infants.

	PNS		Center Adjusted Model		IPTW Model ^1^	
Outcomes	No	Yes	OR (95% CI)	*p*-Value	OR (95% CI)	*p*-Value
Gross motor impairment	171/3114 (5.5)	66/548 (12.1)	2.35 (1.69 to 3.26)	<0.001	1.55 (0.99 to 2.41)	0.053
Cognitive impairment (PARCA score) ^2^	616/2596 (23.7)	158/462 (34.1)	1.66 (1.28 to 2.16)	<0.001	1.08 (0.74 to 1.58)	0.68
Cognitive impairment (ASQ score) ^3^	169/518 (32.6)	29/86 (33.7)	1.06 (0.59 to 1.91)	0.83	1.50 (0.62 to 3.64)	0.35

Values are reported as no./No. (%). ^1^ adjusted for region; ^2^ available for no French centers (*n* = 3058); ^3^ available for French centers (*n* = 604) Descriptive values, OR and *p*-value were calculated after handling missing values by multiple imputation.

## Data Availability

The data generated and analysed in this study are not publicly available. Yet, further enquiries can be directed to the corresponding authors.

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
