# Peer review of "Systemic Steroids in Preventing Bronchopulmonary Dysplasia (BPD): Neurodevelopmental Outcome According to the Risk of BPD in the EPICE Cohort"

_ijerph, 2022, doi:10.3390/ijerph19095600_

Round 1
Reviewer 1 Report
- Page 2, line 82: 36 weeks GA à 36 weeks postmenstrual age
- Page 6, line 205-209: Sentence for hydrocortisone subgroup. The authors stated that cognitive impairment was lower in the hydrocortisone group than in the control group in the high-risk group and in the overall population. Then how were their significances before and after IPTW adjustment? Were they still significant? There is no mention on the statistical significance before and after adjustment.
- Page 6, Figure 2: Figure 3 is misnumbered as Figure 2. In Figure 3, what are the negative numbers in the odds rations and 95% confidence intervals. In general, odds ratios and 95% confidence intervals are positive numbers. Furthermore, why is the format of figure 3 different from figure 2 which is clearly understandable.
- Page 7, line 259-260: The follow-up loss rate is substantial (about 50%). If available, the demographic and clinical characteristics of infants who were lost to follow-up should be provided so that the readers can interpret the results taking into account bias.
- The prediction model of BPD was constructed and applied with the same population. Isn’t there any statistical concern on that?
- How were the rate and severity of BPD of each BPD-risk group (low/moderate/high-risk)? To show the appropriateness of BPD-risk assessment of the authors, these data had better be presented.
- The long-term neurodevelopmental outcome is affected by BPD severity. Did the authors adjust the BPD severity to assess the association between postnatal steroids use and long-term neurodevelopmental outcome?
- The authors did subgroup analysis with infants who were given hydrocortisone. How would it be if subgroup analysis is done with infants who were given dexamethasone, because the association between postnatal systemic steroid and adverse neurodevelopmental outcomes has mostly referred to dexamethasone.
- The timing and duration of postnatal steroids are varying as demonstrated in supplementary table 3. Can you further analyze the association between postnatal steroid and adverse neurodevelopmental outcomes by the timing and duration of postnatal steroids?
- Previous studies reported more harm and less benefit of postnatal steroids in low-risk for BPD infants. The results of this study are not consistent with previous reports. The reason for this discrepancy is worth discussing in the discussion section.
Author Response
Reviewer 1:
- Page 2, line 82: 36 weeks GA à 36 weeks postmenstrual age.
We made the correction, thank you.
- Page 6, line 205-209: Sentence for hydrocortisone subgroup. The authors stated that cognitive impairment was lower in the hydrocortisone group than in the control group in the high-risk group and in the overall population. Then how were their significances before and after IPTW adjustment? Were they still significant? There is no mention on the statistical significance before and after adjustment.
We didn't initially calculate the association before adjustment as we believe that the result after adjustment is the most reliable. I added the result before adjustment in the text (line 228).
- Page 6, Figure 2: Figure 3 is misnumbered as Figure 2.
Thank you for noting this mistake. It has been corrected.
In Figure 3, what are the negative numbers in the odds rations and 95% confidence intervals. In general, odds ratios and 95% confidence intervals are positive numbers. Furthermore, why is the format of figure 3 different from figure 2 which is clearly understandable.
Indeed, the results shown in Figure 3 were the logistic regression coefficients corresponding to the estimated probabilities. We now present the data as Odds Ratio. The new values are presented in Figure 3: the formats of the two Figures have been homogenized as requested. We have also added the results of the Dexamethasone and Betamethasone group (we grouped the 2 molecules because of their similar action and described pharmalogical action). The results of this sensitivity analysis are consistent with the main results on gross motor functions. Note that when the regression coefficient was negative, it indicated a protective effect of hydrocortisone on cognitive development. In the revised version, it becomes an impaired cognitive development in the Dexa/Betamethasone group that is no longer observed in the hydrocortisone group.
- Page 7, line 259-260: The follow-up loss rate is substantial (about 50%). If available, the demographic and clinical characteristics of infants who were lost to follow-up should be provided so that the readers can interpret the results taking into account bias.
Thanks for that comment, it is one the limitation of our study, despite the multiple imputation. We have added a supplemental Table describing the populations “Lost in Follow up” and “Followed up”.
In the cohort, work on loss to follow-up has shown that it is mainly related to social factors and not medical factors. Therefore, simulation analyses in the cohort have shown that estimates of neurodevelopmental impairment are likely underestimated by a small amount, but relationships between variables did not seem to be affected.
We have added a section to the discussion explaining this;
- The prediction model of BPD was constructed and applied with the same population. Isn’t there any statistical concern on that?
We created a model to be able to classify our own cohort in risk groups for BPD. Our aim was not to add another score to the existing ones that have already be been validated. As the only population our score would be used on was the EPICE cohort, we chose not to perform external validation. However, this also means that our score can’t be extrapolated to other populations. However, this was not the purpose of our study. We have clarified this point in the discussion (line 298-300).
- How were the rate and severity of BPD of each BPD-risk group (low/moderate/high-risk)? To show the appropriateness of BPD-risk assessment of the authors, these data had better be presented.
Thank you for this suggestion. We have added the rates of BPD in each group of risk on line 170-171.
- The long-term neurodevelopmental outcome is affected by BPD severity. Did the authors adjust the BPD severity to assess the association between postnatal steroids use and long-term neurodevelopmental outcome?
In the NICHD definition, BPD severity is defined by the respiratory support and the O2 supplementation at 36 weeks PMA, but we didn’t have this data to adjust on BPD severity. In the absence of this information, we adjusted our results on variables associated with the global severity of each patient’s health condition.
- The authors did subgroup analysis with infants who were given hydrocortisone. How would it be if subgroup analysis is done with infants who were given dexamethasone, because the association between postnatal systemic steroid and adverse neurodevelopmental outcomes has mostly referred to dexamethasone.
Thank you for your comment. We have added the results with Dexamethasone and Betamethasone on Figure 3. We found similar results on gross motor development as expected due to the important proportion of patients treated with dexamethasone in the EPICE cohort. But one interesting result is an impact on cognitive development that is no longer observed in the hydrocortisone group. These are secondary analyses and we might lack power to show such a difference, but it raises questions for future research.
- The timing and duration of postnatal steroids are varying as demonstrated in supplementary table 3. Can you further analyze the association between postnatal steroid and adverse neurodevelopmental outcomes by the timing and duration of postnatal steroids?
As we use a large cohort in real life conditions, we unfortunately lack precision on the duration, time of initiation and type of molecule use for each patient. We were therefore not in a position to carry out an in-depth analysis of these parameters and neurodevelopment. However, as the relation between duration and timing of PNS has already been demonstrated, we chose to focus on the modification of this relation by the risk of BPD.
10- Previous studies reported more harm and less benefit of postnatal steroids in low-risk for BPD infants. The results of this study are not consistent with previous reports. The reason for this discrepancy is worth discussing in the discussion section.
The results previously presented were extrapolated from RCT where the risk of BPD was defined by the rate of BPD in the control group. It’s interesting to confront these findings with studies made in a real-life population and using a scoring method applicable in real-life practice. In addition, the fact that no difference was shown in the low risk group might be explained by the small number of treated patients.
Reviewer 2 Report
This article is very interesting and useful for the clinical practice. Congratulations.
I recommend explain the objective in the introduction better.
The abbreviation of neurodevelopmental impairments must specify the first occasion that is named (row 44).
There are two Figures 2 and there aren´t Figure 3, including in the text.
Supplementary table 4. It´s necessary to explain if there are significative differences in the neonatal characteristics between the different treatment (p). Also, I would know the prevalence of association of PNS treatment with two-year neurodevelopmental outcomes according to 200 BPD risk groups with each treatment, not only with hydrocortisone.
In the materials and methods, you write “The patients received either betamethasone, dexamethasone, hydrocortisone or a mixed exposure”. How many patients received mixed exposure? I recommend eliminate the patients with mixed exposure.
Author Response
Reviewer 2:
1- I recommend explain the objective in the introduction better.
Thanks for your comment. We have reformulated the objective to be clearer in the introduction (line 65-69).
2- The abbreviation of neurodevelopmental impairments must specify the first occasion that is named (row 44).
There are two Figures 2 and there aren´t Figure 3, including in the text.
Thank you, we made the appropriate corrections.
3- Supplementary table 4. It´s necessary to explain if there are significative differences in the neonatal characteristics between the different treatment (p).
We have added P values to supplemental Table 5 (former Table 4; numbering has changed because of the addition of a supplemental table on loss to follow-up to the appendix). The only significant difference was on the number of multiple pregnancies.
4- Also, I would know the prevalence of association of PNS treatment with two-year neurodevelopmental outcomes according to 200 BPD risk groups with each treatment, not only with hydrocortisone.
I added the results of the Dexamethasone and Betamethasone group (we grouped the 2 molecules because of their similar action and pharmacological effects). We found similar results on gross motor development which was expected due to the important proportion of patients treated with dexamethasone on the EPICE cohort. But one interesting result is that we show an impact on cognitive development that is no longer observed in the hydrocortisone group. Of course, these are secondary analysis and we might lack power to show such a difference but it’s interesting.
5- In the materials and methods, you write “The patients received either betamethasone, dexamethasone, hydrocortisone or a mixed exposure”. How many patients received mixed exposure? I recommend eliminate the patients with mixed exposure.
There were almost 300 patients who received mixed exposure to PNS. We followed your advice and didn’t include them on the secondary analysis.